# Short-Term Very High Carbohydrate Diet and Gut-Training Have Minor Effects on Gastrointestinal Status and Performance in Highly Trained Endurance Athletes

**DOI:** 10.3390/nu14091929

**Published:** 2022-05-05

**Authors:** Andy J. King, Naroa Etxebarria, Megan L. Ross, Laura Garvican-Lewis, Ida A. Heikura, Alannah K. A. McKay, Nicolin Tee, Sara F. Forbes, Nicole A. Beard, Philo U. Saunders, Avish P. Sharma, Stephanie K. Gaskell, Ricardo J. S. Costa, Louise M. Burke

**Affiliations:** 1Exercise & Nutrition Research Program, The Mary Mackillop Institute for Health Research, Australian Catholic University, Melbourne, VIC 3065, Australia; meganlrross11@gmail.com (M.L.R.); llewis@usada.org (L.G.-L.); iheikura@csipacific.ca (I.A.H.); alannah.mckay@acu.edu.au (A.K.A.M.); louise.burke@acu.edu.au (L.M.B.); 2Research Institute for Sport and Exercise, University of Canberra, Bruce, ACT 2617, Australia; naroa.etxebarria@canberra.edu.au; 3Australian Institute of Sport, Leverrier Street, Canberra, ACT 2617, Australia; nicolin.tee@acu.edu.au (N.T.); philo.saunders@ausport.gov.au (P.U.S.); 4UniSA Online, University of South Australia, Adelaide, SA 5001, Australia; sara.forbes@unisa.edu.au; 5Faculty of Science and Technology, University of Canberra, Bruce, ACT 2617, Australia; nicole.beard1971@gmail.com; 6School of Allied Health Sciences, Griffith University, Gold Coast, QLD 4222, Australia; avish.sharma@vis.org.au; 7Department of Nutrition, Dietetics and Food, Monash University, Notting Hill, VIC 3800, Australia; stephanie.gaskell@monash.edu (S.K.G.); ricardo.costa@monash.edu (R.J.S.C.)

**Keywords:** nutrition, exercise, marathon, athletic performance, gastrointestinal symptoms, intestinal fatty acid binding protein, claudin-3, running, breath hydrogen

## Abstract

We implemented a multi-pronged strategy (MAX) involving chronic (2 weeks high carbohydrate [CHO] diet + gut-training) and acute (CHO loading + 90 g·h^−1^ CHO during exercise) strategies to promote endogenous and exogenous CHO availability, compared with strategies reflecting lower ranges of current guidelines (CON) in two groups of athletes. Nineteen elite male race walkers (MAX: 9; CON:10) undertook a 26 km race-walking session before and after the respective interventions to investigate gastrointestinal function (absorption capacity), integrity (epithelial injury), and symptoms (GIS). We observed considerable individual variability in responses, resulting in a statistically significant (*p* < 0.001) yet likely clinically insignificant increase (Δ 736 pg·mL^−1^) in I-FABP after exercise across all trials, with no significant differences in breath H_2_ across exercise (*p* = 0.970). MAX was associated with increased GIS in the second half of the exercise, especially in upper GIS (*p* < 0.01). Eighteen highly trained male and female distance runners (MAX: 10; CON: 8) then completed a 35 km run (28 km steady-state + 7 km time-trial) supported by either a slightly modified MAX or CON strategy. Inter-individual variability was observed, without major differences in epithelial cell intestinal fatty acid binding protein (I-FABP) or GIS, due to exercise, trial, or group, despite the 3-fold increase in exercise CHO intake in MAX post-intervention. The tight-junction (claudin-3) response decreased in both groups from pre- to post-intervention. Groups achieved a similar performance improvement from pre- to post-intervention (CON = 39 s [95 CI 15–63 s]; MAX = 36 s [13–59 s]; *p* = 0.002). Although this suggests that further increases in CHO availability above current guidelines do not confer additional advantages, limitations in our study execution (e.g., confounding loss of BM in several individuals despite a live-in training camp environment and significant increases in aerobic capacity due to intensified training) may have masked small differences. Therefore, athletes should meet the minimum CHO guidelines for training and competition goals, noting that, with practice, increased CHO intake can be tolerated, and may contribute to performance outcomes.

## 1. Introduction

Elite endurance athletes sustain high-power outputs and extremely high relative exercise intensities, typically well above the crossover point of carbohydrate (CHO) and lipid metabolism. For example, elite marathon runners race at a sustained oxygen (O_2_) uptake of ~85–90% VO_2_max with high rates of substrate turnover [1,2]. Attempts to replace the muscle’s reliance on the relatively finite body CHO stores have been shown to achieve dramatic alterations to muscle substrate use. Indeed, adaptation to low CHO, high fat (LCHF) diets (~65% fat, <2.5 g·kg^−1^·day^−1^ CHO) [3], and more topically, their ketogenic counterparts (~80% fat, <50 g CHO·day^−1^), can double the already enhanced rates of fat oxidation in highly trained endurance athletes to mean maximal rates of ~1.5 g·min^−1^, shifting the intensity at which this occurs from ~45% to ~70% of maximal aerobic capacity [4,5,6]. However, at the intensities at which competitive endurance events are conducted, such increases in fat utilisation do not translate to performance improvements [5,6,7]; indeed, they are harmful to sustained speed or power output, possibly due to the increased oxygen cost associated with fat oxidation (for review, see Burke [8]). Empirical evidence of the reduction in exercise economy associated with adaptation to ketogenic LCHF diets has led to calculations of theoretical advantages to economy and performance following further increases in the contribution of CHO to exercise substrate use [2]. However, our recent investigation of this hypothesis failed to detect a measurable change in oxygen use when elite athletes implemented dietary strategies to increase CHO availability and oxidation during exercise [9]. Nevertheless, there may be other mechanisms by which strategies to increase availability/utilisation of the finite CHO stores might enhance endurance performance.

In order to maximise CHO oxidation during exercise, athletes must both increase endogenous CHO stores via CHO loading and the pre-event meal, as well as maximise exogenous CHO availability by consuming CHO at high rates during exercise [10]. The major rate limiting steps for the oxidation of CHO consumed during exercise involve gastrointestinal issues that dictate CHO transport into the circulation (i.e., feeding tolerance, gastric emptying rate, digestive enzyme secretion and activity from accessory organs and/or villi brush border, intestinal absorption and nutrient transporter enterocyte concentration and activity) [11]. Meanwhile, glucose uptake into the skeletal muscle occurs via GLUT4 transport at the plasma membrane, with ATP production via oxidative pathways being dependent on aerobic adaptations within the muscle (e.g., mitochondrial density), as well as the activation of key enzymes (e.g., the pyruvate dehydrogenase complex) [12]. Contemporary guidelines for CHO intake during endurance/ultra-endurance athletic events recommend the consumption of 60–90 g·h^−1^ [2,13] with a potential dose response for performance in longer events raising interest in strategies to maximise the effectiveness of exogenous CHO oxidation [14]. To facilitate this, it has been proposed that the gastrointestinal tract should be trained by regularly consuming high doses of exogenous CHO during exercise [15,16], targeting adaptations to: (1) increase tolerance to intragastric pressure and enhance gastric emptying [17,18], (2) improve efficient intestinal transit and avoid peristaltic braking mechanisms [19,20], and/or (3) increase CHO transporters at the villi apical brush border to increase absorption capacity [18,19,20]. Although evidence for the latter effect pertains only to the glucose and upregulation of its transport protein, SGLT-1, a similar fructose effect is possible given the rapid upregulation of GLUT-5 to CHO; albeit reported in animal models [21]. The ingestion of artificial sweeteners, namely sucralose, can also increase SGLT-1 content in wild-type mice [22]. The upregulation of intestinal carbohydrate absorption potentially occurs through the stimulation of intestinal nutrient sensing molecules (e.g., T1R3 and α-gustducin), expressed in enteroendocrine cells along the intestinal epithelium, which prompt the mRNA expression and protein synthesis of SGLT-1 through gut hormone regulating pathways (e.g., GIP and GLP-1) [22,23]. These considerations of gastrointestinal enhancement [24] may provide small but meaningful advantages to athletes’ ability to absorb and metabolise exogenous CHO [16,20]. This is particularly important for the scenarios of feeding intolerance, and the compromised gastrointestinal integrity and function associated with exercise stress that are responsible for performance debilitating gastrointestinal symptoms (GIS) [25,26].

Accordingly, the goal of the current study was to compare the effects of a multiple dietary approach to maximise CHO availability (“MAX”) against practices that are recommended to support CHO availability and utilisation during endurance performance [27], but typically implemented at lower levels of the recommended range by real-life athletes [28]. We hypothesised that a combination of strategies to further enhance CHO availability and utilisation during sustained high-intensity exercise would support or improve the gastrointestinal characteristics associated with strenuous exercise, while achieving a performance advantage via enhanced capacity for CHO oxidation at race pace. Specifically, we sought to maximise endogenous CHO contributions through high dietary CHO intake, utilise the most recent evidence to increase exogenous CHO oxidation during exercise, and increase the capacity for exogenous CHO oxidation through “gut-training” with habitual CHO intake during exercise and the regular use of sweeteners. This project was conducted as a sequential investigation of gastrointestinal functional and integrity responses to our protocol in two separate cohorts of high-performance endurance athletes, with the second study also being implemented with the aim of measuring the effect on performance.

## 2. Materials and Methods

### 2.1. Study Overview and Participants 

This study consisted of two very similar and sequential investigations in two different cohorts of highly trained athletes, embedded in highly supervised training camps. This allowed experiences from the first study (data and feedback from the study cohort) to inform small refinements in gut-training elements of the overall intervention for implementation in the second investigation. Both investigations utilised a 14-day dietary intervention, with athletes allocated to either a moderate CHO intake (control: CON) or a novel, multi-pronged experimental approach to increasing CHO availability and utilisation during exercise (MAX). In Study 1, 19 elite male race walkers, ranging from Tier 4/5 (international level competitor, *n* = 15) to Tier 3 (national level training partner, *n* = 4; [29]), were recruited to participate in one of two training camps held in January and May 2018 [9]. In Study 2, 16 Tier 3 runners (11 males, 5 females) were recruited to participate in one of three 3-week training camps held in February, March, and November 2019. Criteria for involvement in the study included completion of the marathon distance (42.2 km) in under 2 h 30 min (males) or 2 h 45 min (females). Two runners participated in two camps and completed both dietary arms of the study, giving a final sample size of 18. All athletes were free of current injury and undertaking consistent training at the time of the camp. Subject characteristics for both athlete cohorts can be found in Table 1, while a summary of the studies is provided in Figure 1.

All camps were held at the Australian Institute of Sport (Canberra, ACT, Australia), where participants either lived in the on-site residences or close by, so that all meals and training could be controlled and mostly completed in a group environment. Prior to, and following the dietary intervention, all participants underwent a series of laboratory and field-based tests to assess body composition, exercise-associated feeding tolerance, and gastrointestinal integrity and symptoms. In the case of the marathon runners, race performance was also assessed. This investigation conformed to the standards set by the Declaration of Helsinki, and each of the studies was approved by the Ethics Committee of the Australian Institute of Sport (20171203 and 20181003 for Study 1 and Study 2, respectively). After comprehensive details of the study protocol were explained to potential participants verbally and in writing, recruited athletes provided their written informed consent.

### 2.2. Overview of Dietary Standardisation

All foods and fluids consumed during the camps in these studies were provided by the Australian Institute of Sport nutrition team. Menu construction and food preparation were undertaken by chefs, food service dietitians, and sports dietitians. Meal plans were individually developed for each athlete, integrating personal nutrition preferences and requirements within their allocated dietary treatment. Meals were eaten in a group setting with individualised menus provided to each athlete. During each meal service, the weight of each food item was recorded using calibrated scales (accurate to 2 g). Individualised snacks were provided for consumption between meals and before training sessions, with their intake being cross-checked at the next meal. A range of energy-free drinks (such as tea or coffee, water) were provided in the participants’ living area with a checklist to allow each participant to report on his/her day’s intake at the first meal of the following day. Participants were prohibited from consuming artificially sweetened drinks or products so that any intake of non-nutritive sweeteners was part of the treatment. During training sessions, CHO-containing drinks and gels were provided according to diet intervention.

### 2.3. Study 1: Race Walking

#### 2.3.1. Overview of Study

An overview of the study which included this dataset is provided in a companion paper [9]. Figure 1 summarises the protocol in which exercise-associated gastrointestinal characteristics were collected; two 26 km race walking sessions involving a hybrid laboratory and field protocol, undertaken pre- and post- a 14-day diet and training intervention.

#### 2.3.2. Overview of Diet and Training Intervention

Full details of this protocol are provided in the companion paper [9]. In brief, the race walkers were allocated to either the moderate CHO diet (CON; *n* = 10) or MAX intervention (*n* = 9) based on athlete preference and the matching of key characteristics including age, body mass, and personal best times. The pre-intervention undertaken by both groups consisted of a 3-day CHO loading diet (8 g·kg BM^−1^·day^−1^), based on the lower end of current sports nutrition recommendation for endurance athletics [2], prior to a ~2-h race walking session. This session was undertaken by a similar standardised approach to CHO availability, provided by a pre-exercise meal (2 g·kg BM^−1^·day^−1^) and CHO intake (30 g·h^−1^) during the session. A 14-day dietary intervention was super-imposed on a supervised, semi-structured training program, inclusive of two long walks (25 km+), an interval training session (10 × 1000 m efforts on a 6 min cycle), and a tempo hill session (14 km, ~450 m elevation) each week. During this period, the CON group consumed a standardised diet which continued to provide CHO availability according to the lower range of the current sports nutrition guidelines (~6.5 g·kg BM^−1^·day^−1^), while the MAX group consumed an iso-energetic diet in which CHO intake was increased in terms of total daily intake (~10 g·kg BM^−1^·day^−1^) and specific intake during exercise (see Figure 1). A daily dose (5 mg·kg BM^−1^·day^−1^) of sucralose in a 2 mM solution was ingested three times per day by MAX group, separate to mealtimes, to maximise absorption. The post-intervention trial was undertaken following 3-day of CHO loading; in the case of the CON group, this protocol was identical to the pre-intervention trial. Meanwhile, the MAX group consumed a CHO loading diet that was higher in CHO (12 g·kg BM^−1^·day^−1^) and based on low-residue (low-fibre) food choices [2], while consuming CHO during the session at higher rates of intake (90 g·h^−1^) (Figure 1). Although there is growing evidence that dietary fermentable oligo-di-mono-saccharide and polyols (FODMAP) affect markers of gastrointestinal function, integrity, and symptoms [30,31,32], we decided not to control FODMAP intake to allow “real-world” dietary habits. We note that tests of body composition and maximal aerobic capacity were undertaken at the commencement and completion of the training camp in which this study was embedded; these results have been reported elsewhere [9].

#### 2.3.3. 26 km Race Walking Protocol and Collection of Gastrointestinal Characteristics

On the morning of each trial, race walkers arrived fasted at the laboratory and filled out a modified visual analogue scale (mVAS) GIS assessment tool [33], which is described in Section 2.6. Breath samples were collected in accordance with clinical gastroenterology guidelines [34], whereby participants were instructed to expire normally twice into a 250 mL breath collection bag that included a mouthpiece and residue bag (Wagner Analysen Technick, Bremen, Germany), with the breath sample being collected on the 2nd expiration. A standardised test meal equating to 2 g·kg BM^−1^ CHO was consumed, followed by a 2 h rest period. Fifteen minutes prior to the start of exercise, venous blood and breath samples were collected, and the GIS tool was administered. Participants then commenced the 26 km race walking protocol, of which detailed methods have been published previously [9]. Briefly, the race walkers performed the first, and every fifth kilometre thereafter, on a motorised treadmill in the laboratory, at either 12 or 13 km·h^−1^, which equated to ~75% VO_2_max and approximated their 50 km race pace. In between each kilometre performed on the treadmill, athletes completed 5 km on a measured outdoor path which surrounded the laboratory at a self-nominated, steady-state speed. Immediately following each treadmill kilometre, the GIS tool was administered, and a breath sample was collected, prior to commencing the outdoor segments. For the final kilometre, a substantial speed increase of 1.3 km·h^−1^ occurred (equating to either 13.3 or 14.3 km·h^−1^) to replicate the top-end finishing speed that typically occurs during a race-walking event [35]. Immediately post-exercise, a venous blood sample was collected, and athletes completed the GIS tool for a final time. Breath samples were collected immediately post-exercise (Ex0), and then at 30 min intervals for the 2 h following exercise, with a final venous blood sample collected 1 h post-exercise (Ex1).

Throughout the outdoor component of the exercise protocol, water was available at ~2 km intervals for ad libitum consumption, as occurs in World Athletics sanctioned race walking events. Athletes were provided with a CHO solution prior to exercise (210 mL) and following each kilometre completed on the treadmill (190 mL), made in-house from glucose, fructose [2:1], and water. During pre-intervention testing, all athletes consumed an 8% CHO solution, providing ~30 g·h^−1^ CHO during exercise. During the post-intervention test, the CON group consumed the same solution, however the MAX group consumed beverages that equated to 24% CHO and provided ~90 g·h^−1^ CHO.

### 2.4. Study 2: Distance Running 

#### 2.4.1. Overview of Study

A similar study involving the CON and MAX interventions was implemented in the cohort of highly trained distance runners (Figure 1) within a 4-week training camp. A field-based running protocol, involving a 35-km run including a 7-km time trial, was undertaken pre- and post-diet training intervention. A total of 18 data sets were collected from the marathon runners (CON, *n* = 8; MAX, *n* = 10). Tests of body composition and maximal aerobic capacity were undertaken at the commencement and completion of the training camp. The multi-session training program outside the laboratory testing and performance protocol included twice-weekly long runs, high intensity running, recovery runs, and gym sessions. All training runs were recorded using a GPS device and heart rate monitor so that all training was accounted for by the research team, and individual diet plans could be adjusted accordingly to support energy needs. While the study intention was for runners to arrive at the camp in near race condition so that improvements in training status would be small, the objective of the training period was to encourage athletes to train hard and maximise the training camp experience.

#### 2.4.2. Body Composition

Body composition assessments were undertaken using DXA according to best practice [36], with the same technician conducting and analysing all scans to reduce variability. Body mass was assessed to the nearest 0.01 kg with digital scales (Seca, Hamburg, Germany). Athletes wore limited clothing and removed all jewellery and metal objects. Body composition (fat-free mass, fat mass and bone mineral content) was assessed using a whole-body scan on a narrowed fan-beam DXA (Lunar Prodigy; GE Healthcare, Madison, WI, USA), with analysis performed using GE Encore 12.30 software (GE Healthcare, Chicago, IL, USA). The DXA technical error of measurement was approximately 0.1% for total mass, 0.4% for total lean, 1.9% for total fat, and 0.7% for total bone mineral content [36].

#### 2.4.3. VO_2_max Testing

Two hours after a standardised breakfast providing 2 g·kg BM^−1^ CHO, runners commenced the VO_2_max testing protocol described previously [9]. Four submaximal running stages, each 4 min in duration and increasing in speed by 1 km·h^−1^ for each subsequent stage, at speeds between 14–19 km·h^−1^, were completed to determine the lactate threshold and turn point. These parameters were used to prescribe target running speeds for the steady-state run and performance test. The VO_2_max protocol consisted of a ramp profile with treadmill speed increasing by 0.5 km·h^−1^ every 30 s for 4 min (to 17 km·h^−1^ and 19 km·h^−1^ for females and males, respectively), after which the gradient was increased by 0.5% every 30 s until exhaustion. Expired gas was collected and analysed using a custom-built indirect calorimetry system described previously [37].

#### 2.4.4. Dietary Interventions

The runners undertook a similar version of the CON and MAX intervention involved in Study 1 (see Figure 1). Individual diets were planned for each participant with a baseline energy intake of ~220 kJ·kg BM^−1^·day^−1^ and the opportunity to manipulate this based on feedback around hunger or BM loss. The pre-intervention race was preceded by 3 days of a CHO loading diet with a CHO intake of 8 g·kg BM^−1^·day^−1^, based on the lower end of the current sports nutrition guidelines for this practice in endurance athletics [2]. During the 14-day dietary intervention, the CON group consumed a standardised diet which continued to provide CHO availability according to the lower range of the current sports nutrition guidelines (~7 g·kg BM^−1^·day^−1^), while the MAX group consumed an iso-energetic diet in which CHO intake was increased in terms of total daily intake (~10 g·kg BM^−1^·day^−1^) and specific intake during exercise (see Figure 1). Protein intake was clamped at 2.2 g·kg BM^−1^·day^−1^ in both conditions, while fat intake was manipulated to make up the remainder of the energy intake. Similar to study 1 [9], dietary strategies to increase CHO content included providing a substantial serve of potatoes as the key CHO source at a meal (e.g., 300 g potato wedges or 500 g mashed potato), increasing the serve sizes of grain products, juices and dairy products at meals and snacks, and increasing training-focused CHO choices (drinks, gels, confectionery). In addition, they consumed the daily dose (5 mg·kg BM^−1^·day^−1^) of sucralose, three times per day, separate to mealtimes. During training sessions, CHO-containing drinks were provided according to diet intervention; CON received 30 g·h^−1^ of a glucose:fructose [2:1] solution (Isoactive, PowerBar, Berkeley, CA, USA), while MAX was required to build up to the highest tolerable rate within the range of 60–90 g·h^−1^ over the period. In a small tweak of the protocol used in Study 1, the MAX group was provided with a CHO hydrogel glucose:fructose sports drink (Maurten, Sweden) due to athlete interest in claims that this product might enhance CHO delivery and mitigate GIS [38]. The post-intervention trial was undertaken following 3-day of CHO loading; in the case of the CON group, this protocol was identical to the pre-intervention trial. Meanwhile, the MAX group consumed a CHO loading diet, higher in CHO (12 g·kg BM^−1^·day^−1^), and based on low residue (low fibre) food choices [2], while consuming CHO during the session at higher rates of intake (60–90 g·h^−1^) (Figure 1). During the post-intervention race, MAX consumed the hydrogel CHO drink at their chosen highest tolerable rates from 70–100 g·h^−1^.

#### 2.4.5. Running Performance

To assess the effectiveness of the dietary intervention on running performance, runners completed an outdoor run of 35 km (5 × 7 km circular laps on flat terrain). This consisted of 28 km of steady-state running at a speed corresponding to 800% of predicted marathon speed, followed by a 7 km self-paced, maximal effort time-trial (TT). To provide an incentive for maximal effort, prize money was awarded to athletes who achieved a TT pace equivalent or greater to 95% of their predicted marathon pace for pre-intervention (TT1), and matched this performance at post-intervention (TT2). Athletes were also ranked and incentivised according to pace above marathon pace in TT1 and improvement in TT2.

Each performance test started at 09:00 following a standardised breakfast providing CHO content of 2 g·kg BM^−1^ and the collection of baseline venous blood and capillary blood samples. Feeding stations were provided every 3.5 km, with CHO solutions individualised according to the diet intervention. Drinks were provided in 125 mL boluses; participants were instructed to consume each drink within 300 m, after which bottles were discarded and weighed to determine exact fluid and CHO intake. At the 21 km mark of the steady-state run (i.e., after lap 3), participants were permitted a 1-min break to change into racing shoes for the last steady-state lap and the 7 km self-paced TT. Total run time, rating of perceived exertion (RPE [39]), and GIS/feeding tolerance were recorded, and blood samples (capillary and venous) were taken immediately, and 1 h post-run (venous samples only). The capillary blood samples were analysed for blood lactate (LactatePro2, Akray, Japan) and glucose (FreeStyle Optium Neo, Abbott Diabetes Care, Doncaster, VIC, Australia) concentrations. To counter any individual differences in the accuracy of these portable analysers, each participant was assigned to a specific device for the duration of their involvement in the study.

### 2.5. Blood Collection and Analysis

Venous blood samples were collected every 5 km during each 26 km walk and pre-, post-, and 1 h post-exercise for the 35 km race. For the race walking protocol, an indwelling cannula (22G Insyte, BD, Macquarie Park, NSW, Australia) was inserted into a vein of the antecubital fossa or forearm after sterilisation of the insertion site using a 70% isopropyl alcohol/2% chlorhexidine swab. For the marathon running race, a vein of the antecubital fossa was sterilised in the same way prior to the insertion of a single-use safety needle (SafetyLok, BD, Macquarie Park, NSW, Australia Australia). Samples were drawn tubes treated with EDTA and/or heparin, and were centrifuged at 1500*g* for 10 min. Supernatant was divided into 1 mL cryotubes and stored at −80 °C until batch analysis could be conducted. Blood concentrations of biomarkers representing damage to intestinal enterocytes and epithelial tight-junction protein integrity were assessed by ELISA. Plasma concentrations of I-FABP (HK406, Hycult Biotech, Uden, The Netherlands) were analysed in both investigations, and claudin-3 (SEF293HU, Cloud-Clone Corp., Katy, TX, USA) was measured only in Study 2. All biomarkers were analysed in duplicate and %CVs were 4.9% for I-FABP and 5.0% for claudin-3.

### 2.6. Gastrointestinal Symptoms (GIS) and Breath H_2_

GIS were measured on 7 occasions throughout the 26 km walk and before and after the 35 km run using a validated and reliable exercise-specific, modified visual analogue (mVAS) scale [33]. This tool assessed gut discomfort, total GIS, upper GIS (i.e., gastro-oesophageal symptoms: belching, heartburn, stomach bloating, upper abdominal pain, urge to regurgitate, and/or actual regurgitation), lower GIS (intestinal symptoms: flatulence, lower abdominal bloating, lower abdominal pain, urge to defecate, and abnormal defecation (e.g., loose watery stools, diarrhoea, and/or faecal blood loss), and other related symptoms that include transient abdominal pain (stitch) and nausea using a 10-point rating scale. Participants were educated and advised to complete the GIS rating scale as follows: 1–4 indicative of mild GIS (i.e., the sensation of GIS, but not substantial enough to interfere with exercise workload) and increasing in magnitude, 5–9 indicative of severe GIS (i.e., GIS substantial enough to interfere with exercise workload), and 10 indicative of extremely severe GIS warranting exercise reduction or cessation. If no GIS were reported by participants, this was recorded as a 0, and subsequently, no GIS severity rating was assessed. Considering GIS, such as regurgitation and defecation, results in the complete or temporary reduction or cessation of exercise, these GIS are presented as 0 and 10 ratings only. Breath samples (20 mL) were analysed in duplicate (coefficient of variation (CV): 3.9%) for H_2_ content using a gas-sensitive analyser (Breathtracker Digital Microlyzer, Quintron, Milwaukee, WI, USA). 

### 2.7. Statistical Analysis

Results are presented as mean values with the associated standard deviation (mean ± SD). Where comparisons between conditions are made, the mean difference with associated confidence limits [95% level] and Cohen’s d effect size is reported. All data were analysed with a general linear mixed model using the R package lme4 (R Studio v4.0.2, R Foundation for Statistical Computing, Vienna, Austria) to accommodate the unbalanced design, missing data, and repeated measurements. For all models, a random effect for participant was included to adjust for baseline levels and interindividual homogeneity. Additionally, sex (runners only), camp and known environmental conditions were accounted for within the model as covariates. For performance time, additional covariates of pre-test body mass and VO_2_max were also included within the model to account for their known influences. Each model was estimated using restricted maximum likelihood, and tests for statistical significance were performed using type II Wald tests with Kenward–Roger degrees of freedom, with Tukey’s post hoc comparisons to detect specific differences. An alpha level of 5% (*p* = 0.05) was used.

## 3. Results

### 3.1. Study 1—Race Walking

#### 3.1.1. Compliance to Dietary Interventions and Exercise Task

The race walkers completed the dietary interventions, with the full results of their intake during each period being reported elsewhere [9]. All participants completed each of the 25 km hybrid laboratory-field sessions. Mean environmental conditions during the outdoor component were 18 ± 4 and 20 ± 2 °C and 41 ± 10 and 52 ± 11% relative humidity for pre- and post-intervention trials for MAX, and 16 ± 4 and 15 ± 5 °C and 49 ± 5 and 60 ± 8% relative humidity for CON, respectively. Although the post-intervention trial in MAX was significantly warmer and more humid than the pre-intervention trial (*p* < 0.05), the differences were small and unlikely to be practically significant.

#### 3.1.2. Gastrointestinal Integrity and Function

Figure 2 summarises the mean changes in plasma I-FABP concentrations before and after the race walking protocol (Figure 2A), observations of individual changes in I-FABP from pre- to post-exercise (Figure 2B), and mean changes in breath H_2_ across the exercise session and during the post-exercise period (Figure 2C). Plasma I-FABP concentration increased from pre- to post-exercise (*p* < 0.001), and remained elevated from pre-exercise at 1 h post-exercise (F(2,85) = 42.8, *p* < 0.001), however no differences between trials (F(1,97) = 2.8, *p* = 0.096) or diets (F(1,18) = 0.185, *p* = 0.672) were evident due to the large individual variability in responses (Figure 2). Furthermore, the mean change in plasma I-FABP from pre- to post-exercise (736 pg·mL^−1^) was considered clinically insignificant. No differences in breath H_2_ between dietary groups were evident across the exercise protocol (F(1,15 = 0.002, *p* = 0.970) or during the post-exercise period (F(1,13) = 0.07, *p* = 0.794), again due to the large individual variability. Within the groups, 5/9 and 4/9 MAX participants registered breath H_2_ concentrations above the clinical threshold for malabsorption (≥10 ppm above basal reading on two consecutive occasions) during the post-exercise period in pre- and post-intervention trials respectively, while the count was 3/10 for the CON group.

#### 3.1.3. Gastrointestinal Symptoms (GIS)

Figure 3 summarises GIS in response to the 26 km race walking protocol, noting peak (Figure 3A), total- (Figure 3B), upper- (Figure 3C), and lower (Figure 3D) GIS separately. Significant 3-way interactions between time, trial, and diets were evident for peak (F(5,184) = 2.9; *p* = 0.014), total-GIS (F(5,184) = 2.7, *p* = 0.023), and upper-GIS (F(5,184) = 2.4; *p* = 0.041, Figure 3). Here, symptoms in the MAX group were greater post-intervention compared to pre-intervention at 13 km, 19 km, and post-exercise (all *p* < 0.01). Additionally, a 2-way diet by trial interaction (F(1,209) = 14.2; *p* < 0.001) was evident for lower-GIS, where symptoms were generally greater during the post-intervention testing compared to pre-intervention in MAX (*p* = 0.001). No differences between trials or time points were evident in CON for any measure of GIS (*p* > 0.05).

### 3.2. Study 2—Marathon Running

#### 3.2.1. Dietary Intakes and Compliance with Interventions

The MAX and CON interventions were completed with good compliance with the intended macronutrient intakes and distribution goals. Table 2 presents the results of energy macronutrient intakes during the three separate phases of the study; the 3-day two loading phases undertaken prior to the 35 km races, and the 14-day intervention diet. Differences between and within groups are noted. In brief, the planned differences were achieved: energy and macronutrient intakes did not differ between CON and MAX groups for the CHO loading protocol for the pre-intervention 35 km running protocol, nor the post-intervention race in the CON group. Meanwhile, the targets for increased CHO for the intervention and CHO loading for the post-intervention 35 km race were achieved ((*p* < 0.001), with a lower fibre intake *p* < 0.001). Protein intakes remained similar between diets and dietary phases.

Table 3 summarises the estimated content of key micronutrients in MAX and CON interventions, noting differences between groups and dietary phases. Daily reference intakes were easily met during the 14-day dietary interventions. In brief, there were small differences between micronutrient content and the density of the two dietary approaches. These included the higher vitamin C (e.g., higher intake of potatoes and fruit) but lower calcium content (e.g., dairy) of MAX vs. CON diet during the intervention phase (*p* < 0.01). The 3-day CHO loading phase of MAX, which deliberately reduced fibre intake via the avoidance of wholegrains and fruit/vegetable skins, was different in several micronutrients (e.g., vitamins B1, B2, B3, folate, calcium) than its CON counterpart (see Table 3).

#### 3.2.2. Body Composition

Changes in body composition are summarised in Figure 4. Pre-intervention BM, fat and lean mass were comparable between groups (*p* > 0.21). Although BM decreased over the 14-day intervention in both (F(1,16) = 7.02; *p* = 0.017), only race day BM showed a significant diet–time interaction (F(1,16) = 5.8; *p* = 0.029, Figure 4A), with a decrease in CON (−0.7 kg, 95 CI −1.1 to −0.3, *p* = 0.043) but no change in MAX (+0.2 kg, 95 CI −0.4 to 0.7, *p* = 0.931). Both groups also lost fat mass (CON = −0.9 kg, 95 CI −1.5 to −0.4; MAX = −0.4 kg, 95 CI –0.6 to 0.1). Here, a significant time:diet interaction effect was seen (F(1,16) = 4.51; *p* = 0.049, ES = 1.03), with a larger reduction in CON (*p* < 0.001), compared to MAX (*p* = 0.268, Figure 4B). No differences between groups (F(1,25) = 0.41; *p* = 0.528), over time (F(1,16) = 0.64; *p* = 0.435) or any interaction (F(1,16) = 0.09; *p* = 0.771) were evident for lean mass.

#### 3.2.3. VO_2_max and Performance

A small and similar increase in VO_2_max (mL·kg^−1^·min^−1^) was evident over the 2-week training period (F(1,12) = 8.4; *p* = 0.013) in both groups (F(1,17) = 0.26; *p* = 0.615) (Figure 4C). Environmental temperature for each of the races were 18 ± 2 °C (race 1) and 19 ± 3 °C (race 2). Mean time for the 7 km TT within the 35 km run at pre-intervention was 1478 ± 131 s [95 CI 561 to 2394] and 1492 ± 130 s [95 CI 567 to 2416] for CON and MAX respectively (Figure 4D). Both groups improved TT performance from pre- to post-intervention (CON = 39 s, 95 CI 15 to 63 s; MAX = 36 s, 95 CI 13 to 59 s; F(1,11) = 15.1; *p* = 0.002). There was no effect of diet on the 7 km TT (ES = 0.08, F(1,15) = 0.006; *p* = 0.941). Significant effects on TT performance were seen for BM (F(1,16) = 7.2; *p* = 0.016) and sex (F(1,14) = 44.1; *p* < 0.001), with males faster than females, and a lighter BM being associated with faster performance times.

#### 3.2.4. Gastrointestinal Integrity and Function

Changes to intestinal epithelial injury (i.e., cellular and tight-junction protein) around the 35 km race are summarised in Figure 5, showing absolute changes (Figure 5A,B) and changes from pre-exercise concentrations (Figure 5C,D) of plasma I-FABP and claudin-3 concentrations. In the pre-intervention trial (Figure 5A), there was a non-significant increase in plasma I-FABP concentrations (ES = 0.56, Figure 5), returning to pre-exercise concentrations at 1 h post-exercise for both dietary groups (F(2,80) = 0.6; *p* = 0.529). Post-intervention, MAX experienced a 4% decrease in concentrations over the exercise protocol compared to a 44% increase in CON, followed by a reduction after 1 h in both groups (CON; ES = 0.36, MAX; ES = 0.27). These differences were not significant, likely due to the large interindividual variability, and were not considered to be of clinical relevance (Figure 5C) (F(2,80) = 0.7; *p* = 0.482) A small yet significant increase in plasma claudin-3 in response to exercise was evident, returning to pre-exercise concentrations 1 h later (F(2,80) = 4.01; *p* = 0.022). A significant trial effect was seen (F(1,80) = 7.8; *p* = 0.007), with a decrease in the claudin-3 response to exercise in the post-intervention trial. However, no interaction with diet was evident (F(1,80) = 3.0; *p* = 0.089).

#### 3.2.5. Gastrointestinal Symptoms (GIS)

Figure 6 summarises the self-reported GIS in response to the 35 km race, noting peak (Figure 6A), total- (Figure 6B), upper- (Figure 6C), and lower- (Figure 6D) GIS separately. Total-GIS increased across exercise (F(1,46) = 11.1; *p* = 0.002, Figure 3), however the magnitude of this increase was similar between trials (F(1,46) = 0.8; *p* = 0.386) and dietary groups (F(1,19) = 3.7; *p* = 0.068, Figure 3). Overall, the CON group experienced a greater magnitude of upper-GIS compared to MAX (F(1,46) = 7.4, *p* = 0.034), with no differences between trials (F(1,46) = 0.3; *p* = 0.611). At pre-intervention, there were no differences between pre- and post-exercise for lower GIS (*p* = 0.394) or peak GIS (*p* = 0.987). Conversely, at post-intervention, significant increases were reported post-exercise for both lower GIS (*p* = 0.016) and peak GIS (*p* = 0.002). No differences between dietary groups were evident for either variable (*p* > 0.05).

## 4. Discussion

We investigated the effects of a novel multi-pronged strategy to maximise endogenous and exogenous CHO availability on gastrointestinal responses and the performance of prolonged strenuous exercise in highly trained endurance athletes in moderate environmental conditions. We compared a diet based on the lower range of the current dietary guidelines for CHO intake for training and competition goals, with a combination of 2 weeks of high CHO support and ‘gut-training’ (daily CHO consumption of 10 g·kg BM^−1^.day^−1^ with the specific practice of in-race CHO intake during workouts and daily use of artificial sweeteners), plus an amplified protocol for CHO loading (3 d of 12 g·kg BM^−1^·day^−1^ from lower residue food sources) and a race intake of 80–90 g·h^−1^. Despite the extreme manipulation of the CHO targets, this short-term dietary periodisation strategy was found to be achievable and well-tolerated. Some markers of gastrointestinal perturbations were increased in response to the first iteration of the within-exercise MAX protocol in the race walkers, and there were continued observations of intra-individual responses in both groups and both diets. Nevertheless, after some modifications of the MAX protocol in the running group, there were no indications of a systematically higher rate of gastrointestinal discomfort or injury, as indicated by responses to the mVAS and circulating levels of I-FABP and claudin-3, with higher CHO intake during sustained high-intensity running mimicking real-life competitive racing. In the case of this latter model, both diets were associated with an improvement in performance at the end of the 2.5 week dietary periodisation, with a failure to detect superior outcomes with the strategies to sustain higher CHO availability and an association between performance improvement and body mass changes across both diets.

The gastrointestinal function makes a dual contribution to sports performance during endurance events, enabling the delivery of nutrients (e.g., CHO) consumed during exercise to move into systemic circulation, to make such nutrients available for skeletal muscle uptake and utilisation, while also having a direct effect on performance according to the absence or presence of GIS [11,16,40]. The risk of the exercise-induced gastrointestinal syndrome and associated GIS during exercise increases with the duration and intensity of exercise [41], as well as the environmental temperature [42,43,44,45], and other extrinsic (e.g., exercise mode and/or pharmaceutical agents) and intrinsic (e.g., biological sex and/or intestinal microbial composition) factors [46]. These factors further interact with the characteristics of food/fluid consumed during exercise (e.g., type, concentration and amount of CHO, total volume of feeding, form, presence of additional ingredients such as caffeine etc.) with the potential to overburden the gastrointestinal tract (for review, see [46]). However, it appears that the gastrointestinal tract has substantial plasticity and can be trained by the habitual practice of the same strategies to better tolerate and absorb CHO intake during exercise [16,20]. Indeed, such gut training (i.e., 2 weeks of 90 g·h^−1^ during 1-h steady-state running) has been shown to lead to better CHO feeding tolerance, availability, and enhanced performance to a gut-challenge protocol, albeit during 2-h steady-state moderate intensity (60% VO_2_ max) running, followed by a 1-h distance test in recreationally trained individuals who were not accustomed to consuming such a high quantity of CHO during exercise [16]. Meanwhile, both meta-analyses [10] and dose-response studies [47] support the performance benefits of higher CHO intakes during exercise. Therefore, a central interest of the current study was whether strategies to acutely increase endogenous and exogenous CHO availability, underpinned by adaptation to a diet providing high CHO availability and specific gut-training, would enhance sports performance at race paces associated with high performance distance events [2], without detriment to gastrointestinal comfort or function. We chose to monitor the gastrointestinal response to standardised race walking and distance running protocols using the combination of a validated GIS assessment tool [33], and carbohydrate malabsorption assessment methods and biomarkers indicative of intestinal epithelial injury.

Overall, there was considerable individual variability in GIS and intestinal epithelial injury (i.e., enterocyte cellular via I-FABP determination and tight-junction protein damage via claudin-3 determination) response to the exercise sessions. These findings are in agreement with the heterogeneity of responses observed within laboratory-controlled exertional and exertional heat stress models [19,44,48,49,50]. Nevertheless, in the race walking arm of the study, athletes following the MAX diet appeared to experience greater severity of some aspects of gastrointestinal dysfunction and GIS in the post-intervention exercise protocol, in which they consumed larger amounts of CHO (90 g·h^−1^) compared to their pre-intervention trial or the protocols followed by the CON group (30 g·h^−1^). For example, they reported greater discomfort associated with upper-GIS (e.g., symptoms synonymous with increased gastric load and intragastric pressure) in the second half of the 26 km walk (~2 h total performance time). Notwithstanding the slightly warmer conditions of the MAX trial day, we redesigned the MAX dietary strategy used in the running study to introduce two small changes with the potential to attenuate GIS, while maintaining the same total CHO. Here, we replaced the exercise CHO source in the MAX treatment (for both gut training and post-intervention trial use) with a commercial CHO hydrogel drink of similar glucose:fructose content. In addition, we altered the frequency of fluid/CHO intake during the performance test (from 190 mL every 5 km/20 min to 125 mL every 3.5 km/12 min). We note that the hydrogel drink is marketed with claims of an enhancement of gastric emptying, gastrointestinal comfort, CHO delivery, and performance [51]. Although these claims were not supported in the earlier literature [38], more recent studies have reported some modest benefits [52,53,54], including enhanced gastric comfort [53] in comparison to traditional CHO drinks, albeit during non-exercise protocols that do not mimic sport [52] or did not use contemporary, validated tools to assess GIS [53]. In the current protocol involving runners, although there were some increases in gastrointestinal discomfort across exercise, there were no differences in the post-intervention trial between the CON and MAX groups, despite the 3-fold increase in the rate of CHO intake in the latter. Indeed, in contrast to the race walking study, upper GIS were lower in the MAX running group compared to the CON group, despite the potential for the greater biomechanical strain of running on the splanchnic arena [55] to exacerbate any GIS.

The investigation of exercise-associated disturbances to the intestinal epithelium via biomarkers has become more popular and sophisticated in recent years [26,56]. Breath H_2_ concentrations, a general measure of carbohydrate malabsorption in the clinical setting [57], have been reported to increase in the recovery period when CHO is consumed before and during exercise. Such results indicate that the load/type of CHO consumed during exercise has overwhelmed the absorptive capacity of the small intestine and/or exercise-associated injury to the intestinal epithelium has resulted in a reduction in absorptive surface or transporters [16,20]. This outcome also manifests in the malabsorption of nutritional beverages consumed in the recovery after intestinal epithelial disturbances caused by high intensity interval exercise [58,59,60], with some subjects achieving results above the clinical threshold of CHO malabsorption [34]. However, a 10–14 day period of ‘gut-training’ with a specific CHO source during exercise was shown to reduce breath H_2_ concentrations in recreational runners who were previously not accustomed to consuming large amounts of CHO during exercise [16,20]. Breath H_2_ concentrations were measured in the race walking protocol of the current study to provide a general measure of the absorption of CHO consumed during the 2-h exercise sessions. While there was a tendency for breath H_2_ to be higher in the post-exercise phase of all trials, the large inter-individual response masked any significant differences between groups or trials. Nevertheless, since the post-intervention trial for the MAX group involved a tripling of the rate of CHO intake compared to all other trials, there is some indication that our overall dietary strategy, which included gut-training, did not cause systematic malabsorption of the larger amounts of CHO. Moreover, considering that the high-level endurance athletes in the current study were generally accustomed to consuming CHO during training and competition, it is likely that they were already somewhat ‘gut-trained’. However, the heterogeneity in CHO malabsorption and GIS response to high CHO dietary intake and CHO intake during our high intensity aerobic race walking and running protocols, including individual breath H_2_ values of clinical significance and severe GIS, highlights the need to personalise CHO intake during training and race nutrition in athletes. The individualisation of sports nutrition strategies is now a prominent theme in sports nutrition practice [27,61].

Changes in splanchnic blood flow and enteric sympathetic activity are normal physiological responses to exercise, known as exercise-induced gastrointestinal syndromes, which nevertheless cause gastrointestinal perturbations via splanchnic hypoperfusion (the circulatory-gastrointestinal pathway) and reduced gastrointestinal functional capacity (the neuroendocrine-gastrointestinal pathway), respectively [43]. Hypoperfusion of the tips of the intestinal epithelial villi may lead to enterocyte necrosis and breaches to the epithelial barrier integrity (e.g., reversible ischemic colitis), with adjunct epithelial hyperpermeability. Such intestinal epithelial status may prompt translocation of luminal pathogenic agents (e.g., bacteria and bacterial endotoxins) into systemic circulation as a result of epithelial cell damage or dysfunction. Plasma concentrations of I-FABP, a biomarker of intestinal epithelial injury that correlates with splanchnic hypoperfusion [62] were monitored in both exercise protocols in the current study. In the race walking protocol, all trials were associated with an increase in I-FABP immediately post exercise, which remained elevated at 1 h into recovery. It is important to note that, despite the substantial exercise stress in the current study, the mean increase in plasma I-FABP concentration across all MAX and CON trials in both studies was modest (~465 pg.mL^−1^). Such responses are in accordance with a protective effect of CHO intake on intestinal epithelium during exercise [62,63,64] and are in contrast to changes of >1000 pg.mL^−1^ seen in response to similar models of exercise stress without CHO consumption during exercise [19,44,45,49,50], which are linked to clinical significance (e.g., warranting management) [27].

Individual variability prevented the detection of differences between groups and trials, but the generally moderate increase confirms that these effects on the gastrointestinal tract are tolerable and recoverable for many athletes. Meanwhile, there were no changes in plasma I-FABP concentrations over the exercise bouts, or differences between groups and trials in the running protocol. Although it is tempting to attribute the apparently lower I-FABP responses in the running protocol to our tweaks of the type and timing of CHO feeding strategies, we note that between study factors are too numerous and future research could unpick this effect further. Notwithstanding the individual variability of I-FABP responses in the participants in both studies, there was no evidence of an increase in intestinal epithelial injury with the MAX intervention despite the substantial increase in CHO intake during the post-intervention trial. This is a valuable finding since although intake of moderate amounts of CHO during exercise is proposed to reduce gastrointestinal injury by increasing splanchnic perfusion [42,43,65], excessive intakes can reverse this effect and exacerbate GIS [11,66]. Finally, our data suggest, that changes in circulating I-FABP concentrations do not correlate with the timing or degree of GIS, which is in accordance with previous laboratory controlled exertional and exertional-heat stress experimental trials [46]. One explanation, at least in the present walking study, is that the major GIS during exercise with CHO ingestion lies in upper-GIS (i.e., gastro-oesophageal symptoms) associated with the neuroendocrine-gastrointestinal pathway of exercise-induced gastrointestinal syndrome, and are generally rapid onset. Meanwhile, increases in plasma I-FABP concentrations and its association with intestinal epithelial cell injury may instigate delayed onset lower-GIS (i.e., intestinal symptoms), in the recovery period, as a result of the circulatory-gastrointestinal pathway. The aetiology of different types of GIS (e.g., rapid or delayed onset, acute or prolonged) has previously been described [41].

The paracellular barrier of the small intestinal epithelium is maintained by several tight junction proteins, including claudin-3. The tight junctions serve as selective barriers that regulate the bidirectional movement of ions, water and other nutrients, while providing protection against the leakage of lumen-originating pathogenic content into the circulation [67]. In the current study, claudin-3, a surrogate marker of tight-junction injury and subsequently intestinal epithelial hyperpermeability, were found to significantly increase with the running protocol and returned to pre-exercise concentrations after 1 h of recovery, with a reduction in this response after the 2 weeks of training in both groups. Again, our study found large inter-individual variability in this biomarker, but no evidence of an increased response to the substantial increase in CHO intake during exercise or following the MAX intervention. Such outcomes are similar to the previous laboratory-controlled experimental trials in which a 3 h running protocol at 2 h 60% VO_2max_ resulted in no change in claudin-3 plasma concentration (~18 ng/mL) [19]. However, any increase in blood concentrations of claudin-3 does not necessarily indicate damage to the tight-junction proteins, and may indicate a systemic response instead [68]. Although the measurement of biomarkers of gastrointestinal permeability, enterocyte, and paracellular stress and injury offers potential, further research is needed to identify best practice protocols and insight around the interpretations of the findings.

The final attribute of the current study was the measurement of performance in the running protocol. Running performance is affected by numerous factors, of which substrate utilisation, exogenous CHO oxidation, exercise economy and GI comfort and can be modified through nutritional strategies. Both groups improved their aerobic capacity over the course of the training phase. However, we hypothesised that in well trained athletes, racing at high relative (of aerobic capacity) intensities, the high CHO availability strategies implemented in the MAX treatment, and particularly on race day, would upregulate CHO oxidation and reduce the O_2_ cost of exercise, leading to improved performance. However, both groups achieved an enhanced TT performance in the post-intervention protocol, indicating a substantial version of the “training camp effect” [7] that may have masked any subtle differences between the MAX and CON treatments. In this regard, we acknowledge several limitations to our performance test (7 km time trial undertaken after 28 km of paced running). By choosing a group race rather than an individual time trial, we introduced the effects of group psychology on the outcome even though this represents ecological validity.

We also note that performance improvements were correlated with a loss of body mass, independent of the dietary treatments. Both groups achieved a small mean loss of body fat and BM over the 14 day training period, with a small regain in BM in the MAX group over the post-intervention CHO loading period, likely due to the increased storage of glycogen and associated water. However, several runners in the CON group experienced a significant loss of body fat which contributed to a difference between the groups. This may have reflected a desire of these individuals to use the training period to manipulate body composition despite our control of diet and training, or the difficulty in achieving energy requirements in this cohort from diets designed with a ceiling on total CHO and protein intake. Indeed, there was a trend to (an unintended) lower energy intake in the CON group which may have reflected the eating behaviour of these individuals. Future studies could focus on a tighter energy match to elucidate the effect of differences in CHO availability, although the effort to achieve such dietary control is noted [69]. For the moment, our study failed to detect performance benefits of training and racing with very high versus high CHO availability, suggesting that CHO intakes reflecting the lower range of the current sports nutrition guidelines are adequate. This adds to the already equivocal literature on “moderate vs. high CHO intakes” in endurance athletes [70] in which we note that differences in performance outcomes across investigations often reflect a range of limitations in study design rather than a true comparison of CHO intake. Such limitations include the choice of a performance outcome which may not be influenced by CHO availability. Furthermore, there has often been a mismatch between CHO intake and CHO availability, with the “moderate” CHO intake treatment being sufficient to meet the fuel requirements of the training load, and therefore fulfilling the description of “high CHO availability”.

In designing the menus for MAX treatment, we deliberately included larger amounts of potatoes and potato-based dishes as an alternative to the focus on grains and sugar-rich processed foods that is typical of the CHO-rich diets chosen by many athletes, especially during CHO loading protocols. The food plans were rated as well tolerated and enjoyable, and were able to easily achieve micronutrient goals, with a few small differences between the micronutrient density of the general CON and MAX menus (e.g., greater Vitamin C for MAX, but greater calcium for CHO). Micronutrient density of the MAX CHO loading protocol (12 g·kg BM^−1^·day^−1^) was altered in comparison to the CON loading equivalent due to the food choices needed to achieve a lower intake of fibre (e.g., reduction in fresh fruit and uncooked vegetables). However, because this is undertaken for such a brief period within the annual training plan of a distance athlete (e.g., 72 h prior to competition peaks, 1–2 times per year), the effect of consuming a diet with micronutrient content less than the population RDI is likely to be minimal. Therefore, another legacy of this study is that a wider range of CHO-rich foods can be used to significantly increase total dietary CHO than is often practiced by athletes. 

The environmental conditions in which elite endurance athletes need to perform are varied and often extreme, including high heat and humidity. This is known to alter GI barrier integrity [41,42] and as such, future research should investigate if the extreme dietary CHO manipulations used in this study are suitable or beneficial in environments that pose a greater threat to physiological function. Our study was also conducted in high-performance athletes, who by nature have a very high degree of metabolic adaptation, potentially including previous familiarisation with high CHO intakes. Therefore, our findings may not apply to lesser trained or lower calibre athletes [29]; this should be understood before such extreme dietary CHO practices are implemented in these groups.

For athletes and sports dietitians who wish to translate the findings from this study into practice, we recommend that high performance endurance events are approached with strategies that at least meet the minimum level of the ranges within recommendations for CHO intake. These should be adapted according to the actual fuel costs of the periodised training programs and competitive events undertaken by each athlete, and their individual experiences of gastrointestinal and performance outcomes. Gut training should also form part of the athlete’s nutritional strategy, particularly when high CHO intakes are planned in race scenarios.

## 5. Conclusions

This study implemented a multi-pronged strategy to increase endogenous and exogenous CHO availability for an acute performance above typical athlete practices and the lower range of current sports nutrition guidelines. The strategy included a 2-week training protocol (including higher CHO intake, regular ‘gut-training’ and daily sucralose loading) as well as aggressive protocols for glycogen supercompensation and CHO intake during an acute exercise task of ~2-h duration. Although there was some evidence of higher gastrointestinal disturbance in the cohort of elite race walkers than with the control intervention, slight tweaking of the dietary protocol in highly trained distance runners resulted in similar/better levels of gastrointestinal comfort than the control group. Measures of biomarkers of GI absorption, intestinal damage and permeability showed high interindividual variability, but no evidence of systematic compromise of gastrointestinal barrier integrity despite intakes of large amounts of CHO during exercise. Despite being well tolerated, the strategy to increase CHO availability during prolonged high-intensity exercise did not translate to a greater improvement in running performance (7 km time trial following 28 km of steady-state running) than seen in the control group. Although this might suggest that further increases do not confer additional advantages above the current guidelines for high CHO availability, it is possible that limitations in our study design (e.g., significant increases in aerobic capacity due to intensified training) and execution (e.g., confounding loss of BM in several individuals despite live-in control in a training camp) may have masked small differences. Further investigation is warranted. Therefore, athletes should meet the minimum CHO guidelines for training and competition goals, noting that, with practice, increased CHO intake can be tolerated and may contribute to performance outcomes.

## Figures and Tables

**Figure 1 nutrients-14-01929-f001:**
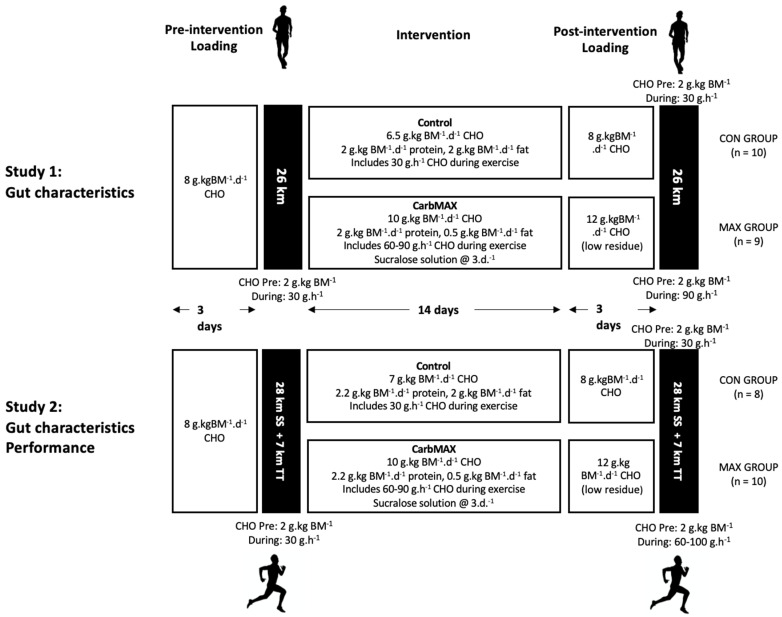
Schematic of the two investigations (race walking: **study 1**) and distance running: (**study 2**) scheduled within a 4-week training camp involving elite male race walkers (*n* = 19) and highly trained marathon runners (*n* = 18). BM: body mass; CHO: carbohydrate; CON: control condition; MAX: carb max intervention; SS: steady-state running; TT: time trial; d: day.

**Figure 2 nutrients-14-01929-f002:**
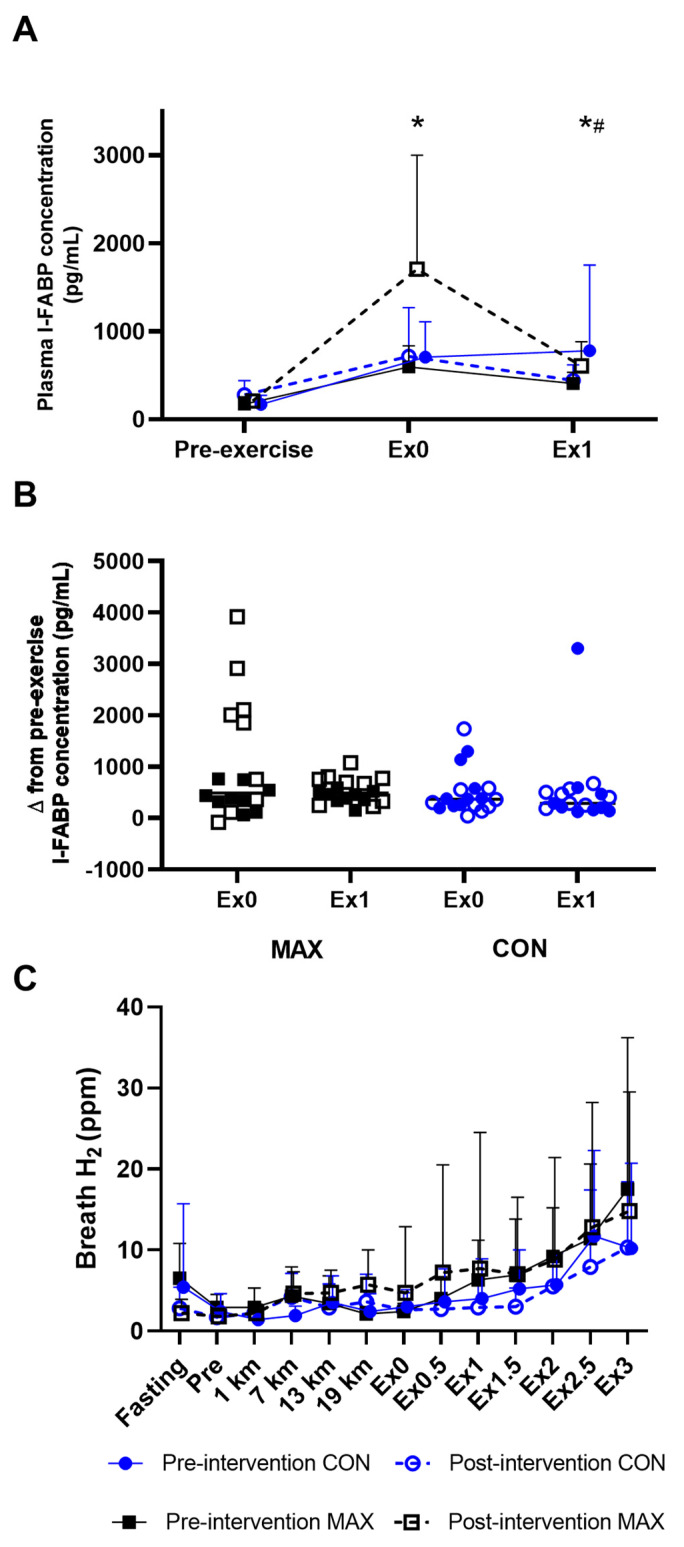
Mean (**A**) and individual changes (**B**) in plasma I-FAPB concentration pre- and post- a 26-km race walking session, and (**C**)mean breath H_2_ values pre-, during, and post-session in elite race walkers before (pre-intervention) and after (post-intervention) a protocol to increase endogenous and exogenous CHO availability (MAX), compared with a control condition (CON) involving moderate CHO availability. Data are mean and standard deviations; * = significant increase from pre-exercise. # = significant decrease from Ex0. Ex0: immediately post-exercise; Ex followed by numeral represents time (in hours) post exercise ending: e.g., Ex1: 1 h post-exercise; Δ (delta): change in reported variable.

**Figure 3 nutrients-14-01929-f003:**
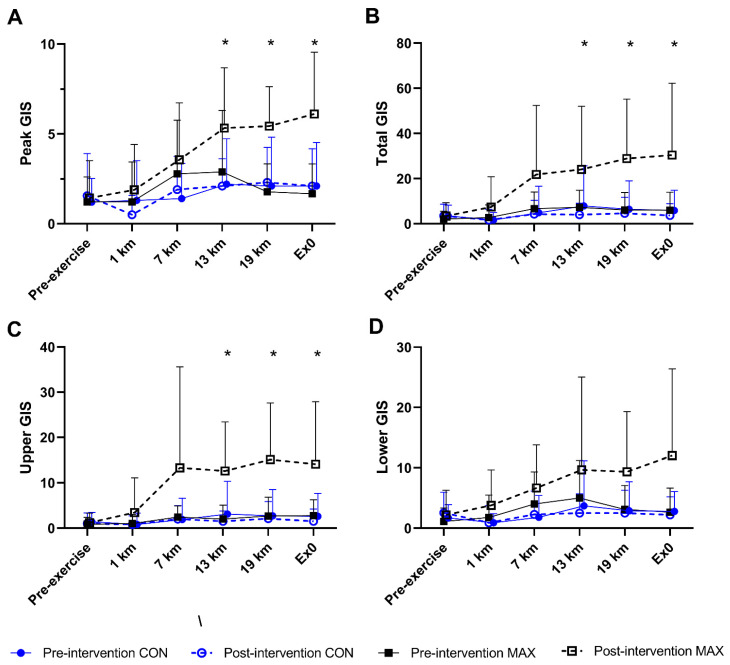
(**A**) Peak, (**B**) total- (**C**) upper-, and (**D**) lower-GIS during a 26 km race walking protocol in elite race walkers before (pre-intervention) and after (post-intervention) a protocol to increase endogenous and exogenous CHO availability (MAX) compared with a control condition (CON) involving moderate CHO availability. Data are presented as mean and standard deviation of summative accumulation of respective GIS category. * = significant difference between the CON and MAX groups.

**Figure 4 nutrients-14-01929-f004:**
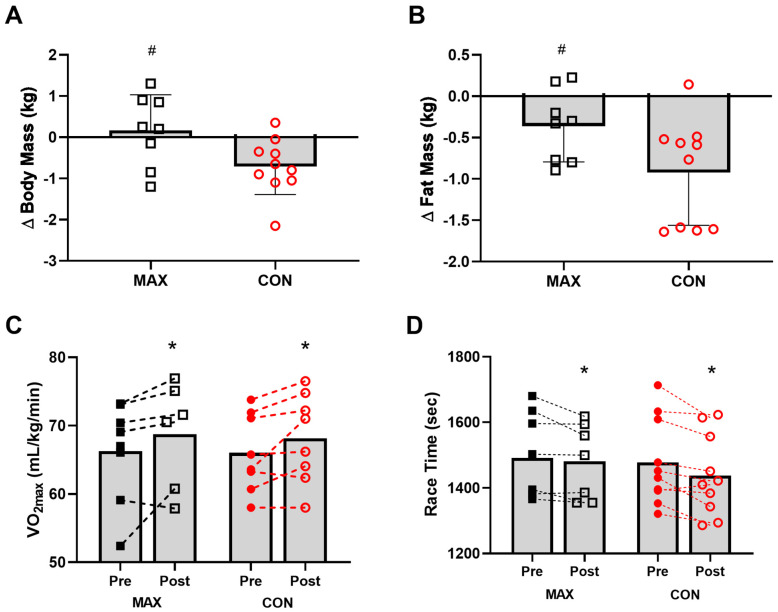
Changes in body mass (kg) (**A**), and fat mass (kg) (**B**) on morning of 35 km race in highly trained marathon runners before (Pre) and after (Post) a protocol to increase endogenous and exogenous CHO availability (MAX) compared with a control condition (CON) involving moderate CHO availability. Relative maximal oxygen uptake (mL·kg^−1^·min^−1^) Pre and Post intervention (**C**) and running performance time (s) for the 7 km time trial (**D**) shown for both groups. Data are presented as mean ± SD. * = significant difference Pre- and Post-intervention; # = significant difference between CON and MAX. Closed squares and circles represent individual data for Pre, open squares and circles represent individual data for Post. Δ = change in reported variable.

**Figure 5 nutrients-14-01929-f005:**
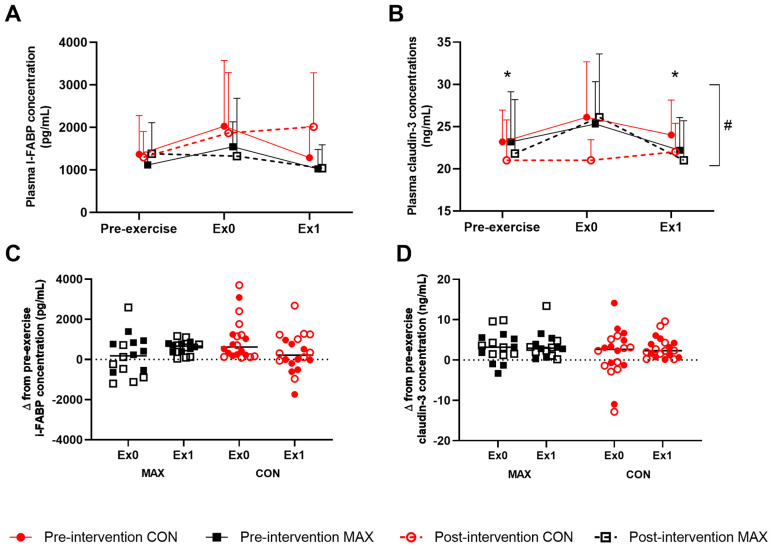
Mean (**A**,**B**) and individual changes (**C**,**D**) in plasma I-FAPB (**A**,**C**) and claudin-3 (**B**,**D**) concentration pre- and/or post-a 35-km race in highly-trained distance runners before (Pre-intervention) and after (Post-intervention) a protocol to increase endogenous and exogenous CHO availability (MAX), compared with a control condition (CON) involving moderate CHO availability. Data are mean and standard deviations; * = significant difference compared to Ex0. # = significant difference between pre- and post-intervention. Δ = change in reported variable.

**Figure 6 nutrients-14-01929-f006:**
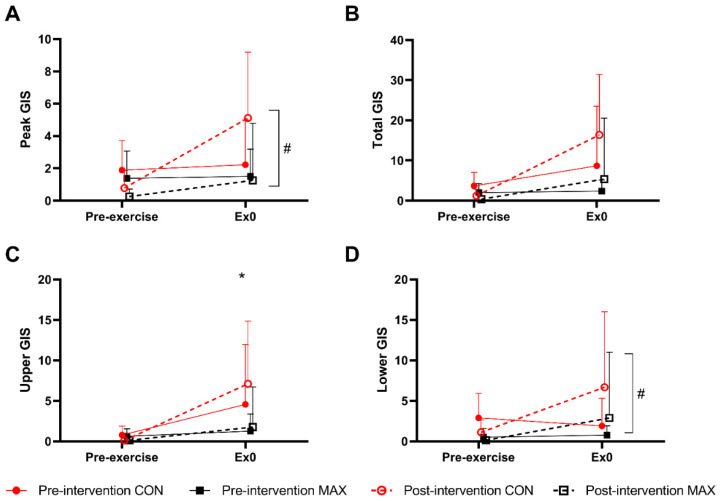
(**A**) Peak, (**B**) total, (**C**) upper, and (**D**) lower gastrointestinal symptoms (GIS) during a 35 km race in highly trained marathon runners before (pre-intervention) and after (post-intervention) a protocol to increase endogenous and exogenous CHO availability (MAX) compared with a control condition (CON) involving moderate CHO availability. Data are presented as means and standard deviations of the summative score for each GIS. * Represents a significant difference between the CON and MAX groups # Indicates a significant difference between pre-intervention and post-intervention.

**Table 1 nutrients-14-01929-t001:** Baseline characteristics of participants.

	Study 1: Race Walkers		Study 2: Marathon Runners	
	CON (*n* = 10 M)	MAX (*n* = 9 M)	CON (*n* = 7 M, 3F)	MAX (*n* = 5 M, 3F)
Age (years)	29.4 (4.6)	29.7 (4.2)	35.1 (7.2)	30.5 (8.1)
Body Mass (BM) (kg)	68.4 (9.4)	68.7 (5.0)	60.6 (8.5)	57.3 (9.7)
VO_2_max (mL·kg^−1^·min^−1^)	60.9 (5.3)	63.1 (4.6)	66.8 (5.8)	65.3 (7.2)
Personal Best (h:min:s)	01:24:30 *	01:23:04 *	02:29:00 ^$^	02:31:00 ^$^
	(0:04:45)	(0:01:59)	(0:06:00)	(0:08:58)

Data are reported as mean (standard deviation); M: male; F: female. Personal best times for 20 km * for 42.2 km ^$^ distance in race walking or running respectively.

**Table 2 nutrients-14-01929-t002:** Energy and macronutrient intakes during pre-race CHO loading phases (pre- and post-interventions) and a 14-day dietary intervention involving high CHO availability (MAX) versus moderate CHO availability (CON) in highly trained distance runners.

			Pre-Intervention Loading						Intervention						Post-Intervention Loading					
			MAX			CON			MAX			CON			MAX			CON		
		Goal Intake	Mean		SD	Mean		SD	Mean		SD	Mean		SD	Mean		SD	Mean		SD
Energy	kJ		13,035	±	2157	13,452	±	1390	13,922	±	2178	13,414	±	1365	15,024	±	2404 ^bbb^	13,757	±	1596
	kJ·kg^−1^		225	±	5.1	223	±	8.7	241	±	21	223	±	16	259	±	11 ^a,bbb^	229	±	30
Carbohydrate	g	Pre: 8 g·kg^−1^ Intervention: MAX: 10 g·kg^−1^ CON: 6.5 g·kg^−1^ Post: MAX: 12 g·kg^−1^	465	±	77	486	±	54	605	±	88 ^aa^	415	±	45	712	±	117 ^aa,bbb^	477	±	55
	g·kg^−1^		8.0	±	0.1	8.1	±	0.2	10.5	±	0.6 ^aa^	6.9	±	0.4	12.3	±	0.4 ^aa,bbb^	7.9	±	0.2
Sugars	g		240	±	43	254	±	33	291	±	42 ^aa^	207	±	23	320	±	74 ^a,bb^	252	±	35
Starch	g		223	±	44	230	±	27	313	±	50 ^aa^	206	±	25	388	±	46 ^aa,bbb^	223	±	29
Protein	g		111	±	21	112	±	9.6	125	±	28	124	±	16	113	±	20	123	±	30
	g·kg^−1^	1.8 g.kg^−1^	1.9	±	0.2	1.9	±	0.1	2.2	±	0.3	2.1	±	0.3	1.9	±	0.2	2.1	±	0.7
Fat	g g·kg^−1^	Pre: 1.5 g·kg^−1^	90 1.5	± ±	15 0.1	93 1.5	± ±	10 0.1	46 0.8	± ±	10 ^aa^ 0.2 ^aa^	117 2	± ±	13 0.2	32 0.6	± ±	7 ^aa,bbb^ 0.1 ^aa,bbb^	99 1.7	± ±	23 0.5
		Intervention:																		
		MAX: 0.5 g·kg^−1^ CON: 1.8 g·kg^−1^																		
		Post: MAX: 0.5 g·kg^−1^																		
Saturated	g		31	±	7	32	±	5	14	±	4 ^aa^	40	±	5	11	±	3 ^aa,bbb^	36	±	14
Monounsat	g		35	±	6	36	±	4	17	±	4 ^aa^	50	±	6	12	±	2 ^aa,bbb^	37	±	6
Polyunsat	g		15	±	3	16	±	1	9	±	2 ^aa^	17	±	3	5	±	1 ^aa,bbb^	16	±	4
Cholesterol	mg		239	±	70	222	±	36	272	±	97 ^a^	379	±	90	191	±	57 ^a^	249	±	51
Fibre	g		47	±	7	47	±	5	50	±	7	48	±	4	29	±	3 ^aa,bbb^	47	±	5
	g·kg^−1^		0.8	±	0.1	0.8	±	0.1	0.9	±	0.1	0.8	±	0.1	0.5	±	0.1 ^aa,bbb^	0.8	±	0.1

^a^ *p* ≤ 0.05, ^aa^ *p* ≤ 0.001 significant difference between diets; ^bb^
*p* ≤ 0.01, ^bbb^ *p* ≤ 0.001 significant difference within a diet between pre- and post-testing.

**Table 3 nutrients-14-01929-t003:** Micronutrient intakes during pre-race CHO loading phases (pre- and post-interventions) and a 14-day dietary intervention involving high CHO availability (MAX) versus moderate CHO availability (CON) in highly trained distance runners.

			Pre-Intervention Loading						Intervention						Post-Intervention Loading					
			MAX			CON			MAX			CON			MAX			CON		
		Goal Intake	Mean		SD	Mean		SD	Mean		SD	Mean		SD	Mean		SD	Mean		SD
Thiamin	mg	RDI: 1.2 mg·d^−1^	2.6	±	0.5	2.6	±	0.3	2.8	±	1	2.7	±	1.2	3.5		0.7 ^bbb^	2.9	±	0.9
	mg/MJ		0.2	±	0	0.2	±	0	0.19	±	0.05	0.19	±	0.07	0.22	±	0.02 ^bb^	0.21	±	0.06
Riboflavin	mg	RDI: 1.3 mg·d^−1^	2.6	±	0.4	2.8	±	0.3	3.3	±	0.8	3.2	±	0.9	3.6	±	0.6 ^bb^	3.2	±	0.7
	mg/MJ		0.2	±	0	0.2	±	0	0.2	±	0	0.2	±	0.1	0.2	±	0.0 ^b^	0.2	±	0
Niacin equivalents	mg	RDI: 16 mg·d^−1^	47	±	8	48	±	5	55	±	16	52	±	13	56	±	10 ^bbb^	51	±	9
	mg/MJ		3.5	±	0.2	3.4	±	0.2	3.8	±	0.6	3.8	±	9.8	3.6	±	0.2	3.6	±	0.5
Vitamin C	mg	RDI: 45 mg·d^−1^	261	±	68	272	±	58	266	±	42 ^aaa^	200	±	28	376	±	173	280	±	46
	mg/MJ		20	±	4	19	±	3	18	±	1 ^aa^	15	±	3	24	±	7	20	±	2
Dietary Folate Equivalents	µg	RDI: 400 µg.d^−1^	1497 111	± ±	291 13	1495 107	± ±	191 7	1310 90	± ±	331 13	1235 89	± ±	274 16	2091 135	± ±	426 ^a,bb^ 18^b^	1665 119	± ±	245 ^b^ 14 ^b^
	µg/MJ																			
Retinol equivalents	µg	RDI: 900 µg.d^−1^	1326	±	210	1381	±	229	1632	±	211	1642	±	248	1386	±	315	1477	±	169
	µg/MJ		100	±	17	100	±	18	114	±	19	121	±	24	91	±	22	106	±	15
Iron	mg	RDI: 8 mg·d^−1^	17	±	3	18	±	2	21	±	3	22	±	2	23	±	4 ^aa,bb^	18	±	2
	mg/MJ		1.3	±	0.1	1.3	±	0.1	1.5	±	0.1	1.6	±	0.2	1.5	±	0.1 ^aa,b^	1.3	±	0.1
Potassium	mg	AI: 3800 mg·d^−1^	5872	±	812	6129	±	598	6250	±	867	5626	±	558	5323	±	837 ^a,bb^	6128	±	512
	mg/MJ		437	±	32	440	±	19	433	±	27	407	±	40	343	±	25 ^aaa,bbb^	437	±	25
Magnesium	mg	RDI: 400 mg·d^−1^	623	±	89	649	±	66	630	±	106	647	±	58	406	±	58 ^aaa,bbb^	652	±	82
	mg/MJ		46	±	2	47	±	2	44	±	5 ^a^	47	±	4	26	±	2 ^aaa,bbb^	46	±	4
Calcium	mg	RDI: 1000 mg·d^−1^	1214	±	201	1354	±	257	1225	±	162 ^aa^	1354	±	139	986	±	178 ^b^	1587	±	852
	mg/MJ		90	±	10	97	±	13	85	±	8 ^aa^	98	±	10	64	±	10 ^aaa,bbb^	102	±	26
Zinc (mg)	mg	RDI: 14 mg·d^−1^	17	±	3	18	±	2	17	±	3	17	±	2	20	±	3 ^bb^	19	±	4
	mg/MJ		1.2	±	0.1	1.3	±	0.1	1.2	±	0	1.2	±	0	1.3	±	0.1	1.3	±	0.1

^a^ *p* ≤ 0.05, ^aa^ *p* ≤ 0.01, ^aaa^ *p* ≤ 0.001 significant difference between diets; ^b^ *p* ≤ 0.05, ^bb^ *p* ≤ 0.01, ^bbb^ *p* ≤ 0.001 significant difference within a diet between pre and post-testing. RDI = recommended dietary intake, d = day

## Data Availability

The datasets for this study are available from the corresponding author upon reasonable request.

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
