# Peer review of "Short-Term Very High Carbohydrate Diet and Gut-Training Have Minor Effects on Gastrointestinal Status and Performance in Highly Trained Endurance Athletes"

_nutrients, 2022, doi:10.3390/nu14091929_

Round 1
Reviewer 1 Report
The abstract is very well written, however, a conclusion sentence should be writen at the end of the abstract, showing thus the main finding of this investigaction, and practical applications of these findings.
The manuscript is very well designed and even though so many data makes difficult to follow the results, the paper touches a very interesting and trendy topic,...with multiple practical aplication that I think they should specify, as suggestions for high performance endurance athletes...
The main objective of this resear should be clearly writtend at the end of introduction section.
This type of research, however, gives rise to further avenues for future research, which should be summarised at the end of the discussion section.
At the same time, while the limitations of the study may help to explain the results obtained, I understand that they should be specified in more detail at the end of the discussion section, and in relation to some of the future research.
Numbering the pages would be a great help in following such a long article.
Author Response
The abstract is very well written, however, a conclusion sentence should be written at the end of the abstract, showing thus the main finding of this investigation, and practical applications of these findings.
We thank the reviewer for their comments and suggestions for the manuscript. We have added a sentence at the end of the abstract and in the conclusion to further define the practical implications. We note that the high CHO intake strategy was associated with improved performance in the current study but did not appear to provide an advantage over practices following the lower range of the current CHO intake guidelines. This may be due to interaction with other factors in the study outcomes that masked any differences. We do note, however, that the higher CHO intakes were tolerated following gut training and were at least compatible with improved performance, if not responsible for it.
“Therefore, athletes should meet the minimum CHO guidelines for training and competition goals, noting that, with practice, increased CHO intake can be tolerated and may contribute to performance outcomes”.
The manuscript is very well designed and even though so many data makes difficult to follow the results, the paper touches a very interesting and trendy topic,...with multiple practical application that I think they should specify, as suggestions for high performance endurance athletes...
We have finalised the conclusion with a sentence of practical recommendation which matches the abstract, as the reviewer suggested. We have also added a short paragraph with some key recommendations following the future research paragraph (lines 768 onwards):
The environmental conditions in which elite endurance athletes need to perform are varied and often extreme, including high heat and humidity. This is known to alter GI barrier integrity [42,43] and as such, future research should investigate if the extreme dietary CHO manipulations used in this study are suitable or beneficial in environments that pose a greater threat to physiological function. Our study was also conducted in high performance athletes, who by nature have a very high degree of metabolic adaptation, potentially including previous familiarisation with high CHO intakes. Therefore, our findings may not apply to lesser trained or lower calibre athletes [74]; this should be understood before such extreme dietary CHO practices are implemented in these groups.
For athletes and sports dietitians who wish to translate the findings from this study into practice, we recommend that high performance endurance events are approached with strategies that at least meet the minimum level of the ranges within recommendations for CHO intake. These should be adapted according to the actual fuel costs of the periodised training programs and competitive events undertaken by each athlete, and their individual experiences of gastrointestinal and performance outcomes. Gut training should also form part of the athlete’s nutritional strategy, particularly when high CHO intakes are planned in race scenarios.
The main objective of this research should be clearly written at the end of introduction section.
Having followed the reviewer’s suggestion, we now believe a thorough list of study objectives is clearly stated in the final paragraph of the introduction (line 104-118). The lengthy objectives section ensures we have captured the multiple aims of the study.
This type of research, however, gives rise to further avenues for future research, which should be summarised at the end of the discussion section.
Thank you for this suggestion. We would note that we have been very honest with our critique of our own study and list several limitations of conducting research that simulates real world practice and race environments. We have, however, added a short discussion involving suggestions for future directions, but have limited this to what we think are the priority areas in the field (line 767).
At the same time, while the limitations of the study may help to explain the results obtained, I understand that they should be specified in more detail at the end of the discussion section, and in relation to some of the future research.
Thank you for this comment. We believe that these concerns are addressed in the manuscript in line with the reviewer’s suggestion. Our fear is that adding further detail would result in repetition and confuse the closing sections of the paper. However, we hope that the addition of the paragraph addressing future research directions also helps to address this comment.
Numbering the pages would be a great help in following such a long article.
We used the ‘Nutrients’ style MDPI template which contains line numbers rather than page numbers, as directed by the journal instructions. (We do agree that future templates could add page numbers as an additional aid to improve readability).
Reviewer 2 Report
The study of King et al aimed to compare the effects of a multiple dietary approach to maximise CHO availability against practices that are recommended to support CHO availability and utilisation during endurance performance. I read the paper with interest and I give some suggestion to the authors
- The inclusion in the article of two studies or two very similar and sequential investigations in two different cohorts makes the text difficult to follow and understand. I recommend remove the first study.
INTRODUCTION
The introduction provides sufficient background information for readers to understand the problem, however, the authors should clarify the importance of training the gut and the inclusion of exogenous CHO. This part would improve this section.
Motivations for this study are more than clear. The objectives are clearly defined at the Introduction and the argumentation in this last part was concise and clarifying.
METHODS
The experimental approach is appropriate for the aim of the study.
This section is well described and allows to replicate the study.
RESULTS
Results paragraphs include more relevant and extended data.
All of the tables include specific, well-developed statistic of the variables analyzed by the authors in the study.
DISCUSSION
Discussion rationale is based on study aims and initial hypotheses. All possible interpretations of the data considered are consistent. But it would be interesting the comparation with other studies with higher CHO intake as well as other population in order to better understand the data obtained.
A practical application section would improve the quality of discussion highlighting the results obtains
Explain limitations of the study an future research lines
The conclusions have coherence with the initial aims, in addition, they are well established and according to the present discussion.
LITERATURE CITED
The literature cited is relevant to the study.
SIGNIFICANCE AND NOVELTY
As it stands, the results are novel and important enough for this journal.
Minor:
The race distance in the abstract and in the document is not the same. Please, be consistent throughout the document.
Author Response
The inclusion in the article of two studies or two very similar and sequential investigations in two different cohorts makes the text difficult to follow and understand. I recommend remove the first study.
We thank the reviewer for their comments and attention to the manuscript. However, we disagree with this suggestion for the following reasons
- Data collected on elite athletes are scarce and important. It is problematic to miss the chance to report the outcomes of the first study
- The two studies were conducted as companion projects with a design that involved similarities but a progression in the design of the intervention (a tweak in the protocol for CHO intake during exercise) and the outcome metrics (progression in the choice of gut parameters as well as the addition of the performance task). As such, there is merit in reporting them as sequential investigations within the same paper
- The similar methodology can be explained once rather than having to be repeated and relearned in two separate papers
- The significance of the progression between the two studies can be explained fully, and placed in context of the practical application to real-world athletic performance
- We are mindful of the temptation to maximise publication outputs by dividing your work into minimal publication units. We are committed to publishing the highest quality papers with maximum data and full interpretation. The journal page allowance supports this, so we have made full use of the opportunity
The introduction provides sufficient background information for readers to understand the problem, however, the authors should clarify the importance of training the gut and the inclusion of exogenous CHO. This part would improve this section.
We have added the word ‘exogenous; to line 86 to reinforce the fact that gut training aims to enhance the capacity to tolerate and absorb CHO sources that are consumed during exercise. Otherwise, we feel the section of this paragraph (line 85-100) is quite thorough in outlining the reported and potential benefits of training the gut in this way. If the reviewer requires more substantial amendments to this effect, could we kindly request they provide more detailed information to where this section does not meet this suggestion currently?
Motivations for this study are more than clear. The objectives are clearly defined at the Introduction and the argumentation in this last part was concise and clarifying.
Thank you. Please see similar comment from reviewer 1, which we have responded to.
Results paragraphs include more relevant and extended data.
Thank you for the feedback on the detail we tried to include
All of the tables include specific, well-developed statistic of the variables analyzed by the authors in the study.
Thank you
Discussion rationale is based on study aims and initial hypotheses. All possible interpretations of the data considered are consistent. But it would be interesting the comparation with other studies with higher CHO intake as well as other population in order to better understand the data obtained.
We agree that comparisons to the literature are an essential framework for data interpretation. However, to our knowledge there are no directly comparable studies in terms of the subject population, i.e. extremely well trained to elite endurance athletes) or the combination of strategies within our intervention. In both the introduction and discussion sections of the paper we have alluded to the scarce number of studies in which gut training has been attempted. However, because of the lack of similarities we have noted them as the background/justification to our study design, and then focussed on the results of our own work. We note that the reviewers have identified the length of the current paper and we are reluctant to further increase it.
A practical application section would improve the quality of discussion highlighting the results obtains
Please see response to a similar comment from reviewer 1, thank you,
Explain limitations of the study an future research lines
In line with reviewer 1 also, this has been included. We feel the limitations of the study have already been thoroughly and honestly interrogated but if the reviewer has specific limitations we haven’t acknowledged we would gratefully receive that feedback.
The conclusions have coherence with the initial aims, in addition, they are well established and according to the present discussion.
Thank you for the feedback
The race distance in the abstract and in the document is not the same. Please, be consistent throughout the document.
Thank you, this has been corrected in the abstract (line 26)